# Paradigm Shift in Gastric Cancer Prevention: Harnessing the Potential of *Aristolochia olivieri* Extract

**DOI:** 10.3390/ijms242116003

**Published:** 2023-11-06

**Authors:** Matteo Micucci, Anna Stella Bartoletti, Fuad O. Abdullah, Sabrina Burattini, Ilaria Versari, Matteo Canale, Federico D’Agostino, Davide Roncarati, Diletta Piatti, Gianni Sagratini, Giovanni Caprioli, Michele Mari, Michele Retini, Irene Faenza, Michela Battistelli, Sara Salucci

**Affiliations:** 1Department of Biomolecular Sciences, University of Urbino “Carlo Bo”, 61029 Urbino, Italy; matteo.micucci@uniurb.it (M.M.); sabrina.burattini@uniurb.it (S.B.); michele.mari@uniurb.it (M.M.); michele.retini@uniurb.it (M.R.); 2Department of Medical and Surgical Sciences, University of Bologna, 40126 Bologna, Italy; anna.bartoletti2@unibo.it; 3Department of Chemistry, College of Science, Salahaddin University, Erbil 44001, Iraq; fuad.abdullah@su.edu.krd; 4Department of Pharmacognosy, Faculty Pharmacy, Tishk International University, Erbil 44001, Iraq; 5Department of Biomedical and NeuroMotor Sciences, University of Bologna, 40126 Bologna, Italy; ilaria.versari2@studio.unibo.it (I.V.); irene.faenza2@unibo.it (I.F.); sara.salucci@unibo.it (S.S.); 6Biosciences Laboratory, IRCCS Istituto Romagnolo per lo Studio dei Tumori (IRST) “Dino Amadori”, 47014 Meldola, Italy; matteo.canale@irst.emr.it; 7Department of Pharmacy and Biotechnology, University of Bologna, 40126 Bologna, Italy; federico.dagostino2@unibo.it (F.D.); davide.roncarati@unibo.it (D.R.); 8Chemistry Interdisciplinary Project, School of Pharmacy, University of Camerino, 62032 Camerino, Italy; diletta.piatti@unicam.it (D.P.); gianni.sagratini@unicam.it (G.S.); giovanni.caprioli@unicam.it (G.C.)

**Keywords:** gastric cancer prevention, nutraceuticals, phytochemicals, *Helicobacter pylori*, *Aristolochia olivieri*, apoptosis

## Abstract

Gastric cancer, particularly adenocarcinoma, is a significant global health concern. Environmental risk factors, such as Helicobacter pylori infection and diet, play a role in its development. This study aimed to characterize the chemical composition and evaluate the in vitro antibacterial and antitumor activities of an *Aristolochia olivieri* Colleg. ex Boiss. Leaves’ methanolic extract (AOME). Additionally, morphological changes in gastric cancer cell lines were analyzed. AOME was analyzed using HPLC-MS/MS, and its antibacterial activity against *H. pylori* was assessed using the broth microdilution method. MIC and MBC values were determined, and positive and negative controls were included in the evaluation. Anticancer effects were assessed through in vitro experiments using AGS, KATO-III, and SNU-1 cancer cell lines. The morphological changes were examined through SEM and TEM analyses. AOME contained several compounds, including caffeic acid, rutin, and hyperoside. The extract displayed significant antimicrobial effects against *H. pylori*, with consistent MIC and MBC values of 3.70 ± 0.09 mg/mL. AOME reduced cell viability in all gastric cancer cells in a dose- and time-dependent manner. Morphological analyses revealed significant ultrastructural changes in all tumor cell lines, suggesting the occurrence of cellular apoptosis. This study demonstrated that AOME possesses antimicrobial activity against *H. pylori* and potent antineoplastic properties in gastric cancer cell lines. AOME holds promise as a natural resource for innovative nutraceutical approaches in gastric cancer management. Further research and in vivo studies are warranted to validate its potential clinical applications.

## 1. Introduction

Gastric cancer includes several subtypes classified by WHO guidelines: adenocarcinoma, ring cell carcinoma, and undifferentiated carcinoma [1].

This pathology represents a leading cause of death globally, with a high incidence in East Asia, Eastern Europe, Central, and South America, and low incidence in other regions, including Italy [2,3].

Approximately 90% of gastric cancer cases are believed to be sporadic and typically develop in individuals aged 45 and above, with a higher incidence in males. Onset below the age of 45 is relatively rare, accounting for about 10% of cases [4].

Environmental factors contributing to gastric cancer risk include *H. pylori* infection, diet, tobacco smoking, and inflammation [5,6].

Chemotherapy and surgery are essential treatments [7], while exploring new preventive measures, including food and vegetal substances inducing apoptosis, holds the potential to strengthen the preventive measures.

Preclinical studies have shown promising results with various food substances [8]. Several phytocomplexes and isolated compounds have demonstrated significant anticancer activities against gastric cancer cell lines. Extracts from hibiscus displayed cytotoxic effects on gastric adenocarcinoma cells [9] without toxic effects on the cardiovascular [10] and central nervous systems [11].

Some phytocomplexes have anticancer properties that, in certain cases, were ascribed to isolated compounds. For instance, an extract from *Patrinia heterophylla* Bunge roots selectively inhibited carcinoma SGC-7901 cells. The active compounds were identified as sarracenin and caffeic acid methyl ester [12]. Moreover, various food components have shown inhibitory effects on cancer cells. Polyphenols such as flavonoids, phenolic acids, stilbens, and diarylheptanoids have demonstrated inhibitory effects towards gastric cancer cells [13]. These findings align with epidemiological studies suggesting that a polyphenol-rich diet may decrease the risk of gastric cancer, particularly in females [14]. Further investigations into these natural compounds may offer promising avenues to design innovative chemopreventive strategies in gastric cancer.

In our study, we chose *Aristolochia olivieri* Colleg. ex Boiss. due to its historical use in Kurdish folk medicine for gastrointestinal ailments. We opted for a methanolic extract because methanol is an effective solvent for extracting phenolic acids and flavonoids, compounds with known potential against gastric cancer. This choice allowed us to comprehensively examine the plant’s chemical composition and evaluate its anti-cancer properties, making it a well-suited candidate for addressing gastric cancer.

Indeed, we conducted a comprehensive evaluation of the chemical composition and in vitro antibacterial and anticancer effects of a methanolic extract obtained from the aerial parts of *Aristolochia Olivieri* Colleg. ex Boiss., a plant used in Kurdish folk medicine for gastro-intestinal ailments. The results revealed the presence of several phenolic acids and flavonoids. The experiments demonstrated significant anticancer activities through apoptosis. These findings highlight the potential of AOME as a novel resource for developing innovative approaches in the management of gastric cancer.

## 2. Results

### 2.1. AOME Chemical Characterization

AOME exhibited diverse compounds, categorized into flavonoids and phenolic acids, measured in mg kg^−1^. Among the flavonoids, delphinidin-3-galactoside had the highest concentration (19.85), followed by rutin (483.97), hyperoside (2540.33), isoquercitrin (986.27), delphinidin-3,5-diglucoside (867.66), and kaempferol-3-glucoside (342.52). Notably, other polyphenols like (+)-catechin, procyanidin B2, (−)-epicatechin, cyanidin-3-glucoside, petunidin-3-glucoside, quercitrin, myricetin, naringin, hesperidin, phloretin, kaempferol, and isorhamnetin were not detected.

Among phenolic acids, caffeic acid displayed the highest concentration (9867.16), followed by p-hydroxybenzoic acid (725.76), trans-cinnamic acid (603.99), gallic acid (17.07), neochlorogenic acid (14.49), 3-hydroxybenzoic acid, vanillic acid, resveratrol, syringic acid and procyanidin A2 were not detected.

The total content of identified polyphenols in AOME was 30956.87 mg kg^−1^ (Table 1).

### 2.2. Antibacterial Activity

The AOME’s antimicrobial activity against Helicobacter pylori was investigated. DMSO was used as a negative control. DMSO did not have any significant impact on Helicobacter pylori viability.

On the other hand, AOME showed antimicrobial effects against Helicobacter pylori. AOME MIC and MBC values were 3.70 ± 0.09 mg/mL. These two values overlapped, indicating the phytocomplex’s ability to exert a cytotoxic effect at the tested concentrations.

These findings suggest that AOME possesses antimicrobial activity against Helicobacter pylori. The MIC and MBC values demonstrate AOME’s in vitro efficacy at relatively high concentrations.

### 2.3. The Antineoplastic Properties

The impact of AOME on AGS and KATO III cell viability was assessed in a dose- and time-dependent manner (Figure 1, graphs 1 and 2). The response of SNU1 cells to AOME (graph 3) differed from the other cell lines. SNU1 cells displayed resistance to AOME within the initial 24 h of treatment; however, their viability declined significantly after 48 h of treatment, and cell death became evident upon AOME administration at its bactericidal dosage. The dosage of 7.4 mg/mL was excluded due to complete cell susceptibility.

To further elucidate the effects of AOME on cell proliferation and survival over extended periods, KATO and AGS cells were exposed to increasing concentrations of AOME for 48 and 72 h. The treatment significantly affected cell proliferation and viability.

The optimal cell treatment time for the apoptotic effects was 48 h. KATO III cells showed a lower IC_50_ at 72 h. IC_50_ values for AGS and KATO III cell lines at 48 h were about 1.34 mg/mL and 1.28 mg/mL, respectively, and at 72 h were 1.45 mg/mL and 0.939 mg/mL, respectively (Figure 2).

### 2.4. Ultrastructural Analyses

Morphological analyses confirmed the cytotoxic effect of AOME on gastric cancer cells, exhibiting a dose- and time-dependent response. However, the impact varied among the cell lines, particularly with SNU 1 showing greater resistance to AOME treatment (Figure 3).

AGS control demonstrated well-preserved proliferating cells (A–F), as evidenced by the number of metaphases (A). Cells appeared elongated with prominent nucleoli and diffuse chromatin in the nuclei (E, F). SEM images (B) showed intact cellular membranes. TEM analysis revealed well-preserved mitochondrial structures (E, inset F).

At lower and medium doses of AOME, metaphase cells decreased (G, L), but the cells maintained an elongated morphology. SEM showed the formation of blebs (H, I) at both concentrations. In the case of low concentration, the blebs were associated with autophagic vacuoles in the cytoplasm (K), while at medium concentration, micronuclei and chromatin condensation were observed (R, S). At medium doses, a small percentage of cells showed autophagic vacuoles and disrupted mitochondria (inset R).

At the bactericidal dose, rounding cells indicative of cytoskeleton rearrangement were observed (P), along with numerous blebs on the cell surface and a discontinuous membrane (M, Q). TEM images showed chromatin condensation typical of apoptotic cells, as well as micronuclei ejected from the cells, suggesting secondary necrosis (R, S).

Kato III cells (Figure 4) appeared to be elongated under control conditions (A, D–F), forming monolayers joined by junctions (E). The nuclear membrane was well preserved, although rare small blebs were occasionally observed (C, D). Cells gradually exhibited a rounding phenotype (G, M, P), becoming spherical at the bactericidal dose (S). Blebs appeared at all AOME concentrations (H, I, N, O, U) with higher doses showing discontinuity in the cell membrane (T). Numerous autophagic vacuoles were observed at low and medium doses (K, Q), accompanied by mitochondrial (R) damage and chromatin condensation (V). At high doses, many cells exhibited morphological features typical of secondary necrosis (W, Z).

SNU 1 cells (Figure 5) displayed a rounded morphology under control conditions (A, E, L, P) with well-preserved mitochondria (C). At low doses, numerous autophagic vacuoles were observed (H, I). Several membrane blebbing appeared at all concentrations (F, M, Q). Bactericidal dose induced membrane damage (Q), nuclear membrane detachment, and chromatin condensation (N, O, R, S).

## 3. Discussion

Gastric cancer is a global health concern. Its development is influenced by various factors, including genetic predisposition, environmental exposures, diet, and microbial infections [1].

The “exposome” encompasses life-long environmental exposures: diet, lifestyle, pollutants, and infectious agents. Helicobacter pylori infection is a major risk factor for gastric cancer, affecting over half the population [5]. Currently, investigating the potential anti *H. pylori* and anticancer properties of food and vegetal substances, such as AOME, represents a promising avenue for developing preventive strategies for gastric cancer prevention.

AOME chemical composition differs from other plants of the same genus, with varying concentrations of compounds across different species. AOME contains moderate concentrations of compounds like gallic acid, neochlorogenic acid, and delphindin-3-galactoside, ranging from 14.49 to 19.85 mg kg^−1^.

However, several compounds such as (+)-catechin, procyanidin B2, (−)-epicatechin, cyanidin-3-glucoside, and petunidin-3-glucoside were not detected. On the other hand, AOME exhibits significant concentrations of p-hydroxybenzoic acid (725.76 mg kg^−1^), 3-hydroxybenzoic acid (1160.66 mg kg^−1^), caffeic acid (9867.16 mg kg^−1^), and p-coumaric acid (9976.83 mg kg^−1^).

In our study, we explored the antibacterial and anticancer activities of AOME. The antibacterial activity against *H. pylori* occurred at a relatively high concentration (MIC and MBC values of 3.70 ± 0.09 mg/mL). As the high concentration makes it challenging to apply AOME directly as antibacterial agent, we aimed to investigate the effect of AOME’s in vitro effects against gastric cancer.

We observed significant alterations in cellular morphology, indicative of apoptotic effects. These alterations included prominent vacuolization and changes in mitochondrial structure, as well as evident signs of autophagy and mitophagy.

Observed vacuolization implies intracellular vesicle formation, linked to cellular stress and apoptosis pathways. Changes in mitochondrial morphology align with apoptotic cell death, indicating mitochondrial dysfunction [15]. Furthermore, the presence of autophagic vacuoles and mitophagic structures indicates that AOME treatment may trigger cellular self-degradation processes, suggesting a potential involvement of autophagy in apoptotic effects. These findings align with the growing evidence of crosstalk between apoptosis and autophagy pathways, where autophagy can either promote or suppress apoptosis depending on the cellular context [16]. It is important to highlight that our study focused on morphological changes, rather than on investigating the specific molecular mechanisms underlying these effects. While we did not delve into the detailed molecular pathways, the observed alterations in cellular morphology provide valuable preliminary insights into AOME’s potential role in inducing apoptosis in gastric cancer cells.

Comparing the IC_50_ values of individual compounds reported in the literature, with their concentrations at the IC_50_ value of AOME at 48 h, caffeic acid, rutin, and quercetin may contribute to the observed effect of AOME.

Ultrastructural changes, including blebs formation, chromatin margination, and apoptotic bodies, support the apoptotic effects of AOME polyphenols. Indeed, apoptotic blebs are membrane protrusions occurring during apoptosis, induced by caffeic acid, quercetin, and rutin acting through a multitarget mechanism [17,18,19]. These three polyphenols share some key molecular pathways underlying the induction of apoptosis.

Specifically, these compounds promote apoptotic bleb formation, as they affect the expression of pro-apoptotic and anti-apoptotic proteins such as Bad, Bax, Mcl-1, and Bcl-2, leading to a delicate balance shift tipping the scales towards apoptosis in gastric cancer cells.

Additionally, they promote the production of ROS and ΔΨm. Mitochondrial dysfunction and loss of ΔΨm are key events in the apoptotic process, leading to the release of pro-apoptotic factors and the initiation of blebbing [20].

The significant alterations in cellular morphology due to AOME may result from the synergistic interaction among the various phytochemicals.

In summary, apoptotic blebs formation represents a morphological hallmark of the apoptotic process triggered by these compounds.

The other compounds were detected at low concentrations; however, we cannot completely exclude their contribution to AOME’s apoptotic effects. For instance, ellagic acid [21] p-coumaric acid, gallic acid [22], and quercetin [19] may act through complementary mechanisms, targeting different pathways involved in cell survival and the proliferation of AGS cells.

Ellagic acid inhibits cell migration and induces apoptosis [21]. Gallic acid triggers apoptosis through the intrinsic pathway, involving the activation of caspases, up-regulation of pro-apoptotic proteins, and down-regulation of anti-apoptotic Bcl-2 family proteins. It also induces the expression of death receptors (Fas, FasL, and DR5) mediated by p53 [23]. Quercetin induces cell apoptosis in AGS cells by increasing ROS production, reducing ΔΨm, and altering the expression of apoptosis-related genes [19]. This complementary action of quercetin in inducing apoptosis via ROS-mediated pathways, along with ellagic acid and gallic acid’s effects on gene expressions and intrinsic apoptotic pathways, may offer a multifaceted approach to combat gastric cancer by targeting various aspects of the disease.

Ellagic and gallic acid induce apoptosis through different pathways: ellagic acid alters gene expression related to apoptosis, migration, and inflammation, while gallic acid triggers the intrinsic pathway via caspases and Bcl-2 family proteins. Quercetin, on the other hand, inhibits cells growth.

Synergy occurs when the combined effect of multiple compounds is greater than the sum of their individual effects. The presence of diverse bioactive compounds in the AOME extract could potentially create a synergistic network, where the interactions between different compounds amplify their overall cytotoxic and apoptotic effects, thereby increasing the potency of each molecule.

Overall, the observed apoptotic activity of the AOME extract in gastric cancer cells, despite many compounds being present at subactive concentrations, suggests a complex interplay and synergistic effects among the bioactive compounds.

Our preliminary findings suggest that AOME may hold promise in a distinct clinical space, which may not be primarily associated with its antibiotic effects, but rather with the administration of the phyto-complex to individuals who have contracted *H. pylori* infection and, as a result, are at a higher risk of developing gastric cancer. This perspective aligns seamlessly with the concept of nutraceuticals before drugs beyond foods.

## 4. Material and Methods

### 4.1. Plant Material

The *Aristolochia olivieri* Colleg. ex Boiss. plant was collected in April from Sakran Mountain, northern Iraq. The voucher specimens, accession number (7705), were deposited in the Herbarium at Salahaddin University-Erbil, Iraq. The leaves were air-dried separately, under shade, at room temperature (20–25 °C).

### 4.2. Preparation of Extract

The leaves of the *Aristolochia olivieri* Colleg. ex Boiss. were dried for 48 h in an oven at 40 °C. The dried plant material was then crushed and pulverized. Subsequently, 40 g of this material were extracted using pure methanol (750 mL) in a Soxhlet apparatus (DWK Life Sciences Kimble™ KIMAX™ Soxhlet Apparatus with Allihn Condenser, Thermo Fisher Scientific Inc., Waltham, MA, USA) for a period of 24 h. The extract volume was concentrated using a rotary vacuum evaporator at a temperature of 40 °C until it formed a powder, which was collected and used for further experiments.

### 4.3. Reagents and Standards

Cyanidin-3-glucoside chloride, delphinidin-3,5-diglucoside chloride, delphinidin-3-galactoside chloride, petunidin-3-glucoside chloride, malvidin-3-galactoside chloride, quercetin-3-glucoside, and kaempferol-3-glucoside were purchased from PhytoLab (Vestenbergsgreuth, Germany). The remaining 31 analytical standards of the 38 phenolic compounds were supplied by Sigma-Aldrich (Milan, Italy).

### 4.4. Analytical Characterization of AOME

Stock solutions (1000 mg L−1) of analytes were prepared in HPLC-grade methanol and stored at 4 °C (or −15 °C for anthocyanins). Daily, working solutions were made by dilution. Formic acid (99%) from Merck (Rahway, NJ, USA), hydrochloric acid (37%) from Carlo Erba Reagents (Emmendingen, Germany), HPLC-grade methanol from Sigma-Aldrich, and deionized water (>18 MΩ cm resistivity) were purified using a Milli-Q SP Reagent Water System (Merck Millipore, Burlington, MA, USA). All solvents and solutions were filtered through a 0.2 μm polyamide filter from Sartorius Stedim (Göttingen, Germany). Samples were further filtered with Phenex™ RC 4 mm 0.2 μm syringeless filter, (Phenomenex, Torrance, CA, USA), before HPLC analysis.

### 4.5. HPLC-ESI-MS/MS

HPLC-MS/MS analysis was conducted as previously reported [24].

### 4.6. Cell Culture and Treatments

The three commercial ATCC GC cell lines, AGS (ATCC-CRL1739), KATO-III (ATCC-CRL5971), and SNU-1 (ATCC-HTB103), were cultured as previously reported [25,26,27]. These cell lines represent distinct histological subtypes of gastric carcinoma. AGS models adenocarcinoma, KATO-III represents signet ring cell carcinoma, and SNU-1 corresponds to undifferentiated adenocarcinoma. They are widely employed in scientific research for their ability to cover the spectrum of gastric carcinoma histologies, making them valuable tools for validation studies in the field. All cell lines were exposed to AOME dissolved in DMSO at different concentrations (7.4 mg/mL, 3.7 mg/mL, 1.9 mg/mL, and 0.9 mg/mL). Effects were evaluated after 24 h, 48 h, and 72 h using cell viability, proliferation tests, and ultrastructural investigations.

### 4.7. Trypan Blue Dye Exclusion Assay

The effect of AOME on cell viability was determined using Trypan blue exclusion and the count of live and dead cells performed with a hemocytometer. Briefly, cells were washed and suspended (1.0 × 105 cells/mL) in a solution of 1× PBS: 0.5 mM EDTA (Life Technologies, Carlsbad, CA, USA) containing 0.2% BSA (Sigma-Aldrich). Fifty microliters (50) of the cell suspension was taken and mixed with an equal volume of 0.4% Trypan blue. The solution was mixed thoroughly and allowed to stand for 5 min at room temperature. Ten (10) microliters of the solution was transferred to a hemocytometer, and both viable (clear) and dead (blue) cells were counted. The number of live cells divided by the total number of counted cells (clear and blue) gave the percentage viability [25]. Viability graphs have been realized with Graph Pad Software (version 10.1.0 (316), Graphpad Software Inc., La Jolla, CA, USA) and, for statistical analysis, *t*-test has been performed, evaluating control (ctrl) sample vs. treated sample both at 24 h and 48 h of treatment. Data have been considered statistically significant with a *p*-value <0.05 (asterisk) and highly significant with *p*-value <0.005 (two asterisks) [28].

### 4.8. Cell Proliferation Assay

Cells were seeded in 96-well plates (5000 cells/well, 100 µL medium) and incubated for 24 h for adherence. Kato and AGS gastric cancer cell lines were then cultured with vehicle (DMSO 0.1%) or increasing AOME concentrations (0.47–3 mg/mL) for 48 and 72 h [28]. Cell growth was measured using PrestoBlue reagent (Invitrogen, Monza, Italy). After adding PrestoBlue (10 µL) and incubating for 2 h, absorbance was read at 600 nm using an Infinite M200 photometer (Tecan Group Ltd., Männedorf, Switzerland). Three replicates per concentration were tested, with at least three independent experiments (bars, ±s.d.).

### 4.9. SEM

Different gastric cells, after being washed in 0.1 M phosphate buffer, were fixed in suspension with 2.5% glutaraldehyde. Afterwards, they were deposited on poly-L-lysine-coated coverslips overnight at 4 °C and processed as previously described [29].

### 4.10. TEM

AGS and KATO III cells were washed and fixed with 2.5% glutaraldehyde in 0.1 MPBS for 30 min, then scraped and centrifuged at 300× *g* for 10 min. The pellets were fixed in 2.5% glutaraldehyde for an additional 30 min. SNU1 cancer cell pellets were immediately fixed in 2.5% glutaraldehyde in 0.1 M in phosphate buffer (pH 7.3). All cells were post-fixed in 1% OsO4 for 1 h, alcohol dehydrated, and embedded in araldite. Thin sections were stained and analyzed using a Philips CM10 transmission electron microscope (Philips, Amsterdam, The Netherlands) [29].

### 4.11. Antibacterial Activity

AOME’s antimicrobial activity against Helicobacter pylori was evaluated following the CLSI guidelines with slight modifications. MIC and MBC were determined using the broth microdilution method. Extracts were prepared from the powder in sterile distilled water and twofold diluted in a 96-well microtiter plate (final concentrations: 0.19–25 mg/mL). *H. pylori* liquid culture was added, and plates were incubated for 72 h at 37 °C under microaerophilic conditions. MIC was the lowest concentration inhibiting visible bacterial growth, while MBC was determined by plating diluted cultures and incubating at 37 °C for 48 h under microaerophilic conditions. Negative and positive controls were included. The MIC (Minimum Inhibitory Concentration) and MBC (Minimum Bactericidal Concentration) values were determined based on three replicates of each test. These values were assessed descriptively, as the primary analysis focused on identifying the lowest concentration that inhibited bacterial growth (MIC) and the concentration that resulted in bacterial cell death (MBC) on established microbiological guidelines and the consistency of results across the repeated experiments [30].

### 4.12. Statistical Analysis

Statistical analysis, including one-way ANOVA and post-hoc Tukey’s HSD tests (*p* < 0.05), was applied to both the Trypan Blue Dye Exclusion Assay and the Cell Proliferation Assay data to identify significant differences among AOME concentration groups.

## 5. Conclusions

In conclusion, AOME exhibits antimicrobial activity against *H. pylori* and potent antineoplastic properties in gastric cancer cell lines, showing promise as a natural resource for innovative nutraceutical approaches in gastric cancer management.

Further research is warranted to elucidate the precise mechanisms of action and synergistic interactions within the AOME, which could pave the way for the development of novel nutraceutical strategies contributing to gastric cancer prevention.

## Figures and Tables

**Figure 1 ijms-24-16003-f001:**
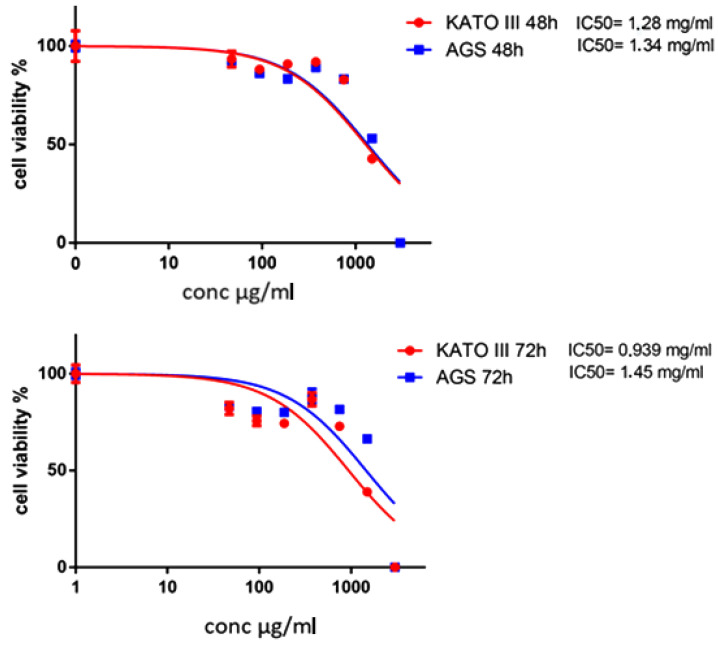
AOME’s impact on the viability of KATO III and AGS cell lines at 24 and 48 h of incubation.

**Figure 2 ijms-24-16003-f002:**
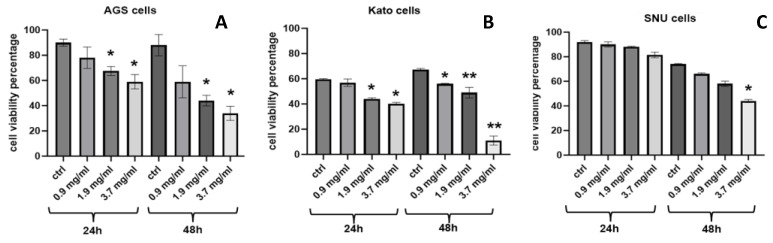
Effects of AOME on Gastric Cancer Cells: In vitro investigation of AOME’s impact on the proliferation and apoptosis of AGS, KATO III, and SNU1 cells. Gastric cancer cells were treated with AOME at concentrations of 0.9, 1.9, and 3.7 mg/mL for 24 or 48 h. (**A**) Colony Formation Assays were performed to assess the proliferation of AOME-treated cells, (**B**,**C**) the data presented represent the mean ± SD of two independent in vitro experiments, we indicate the more significant result with * and **.

**Figure 3 ijms-24-16003-f003:**
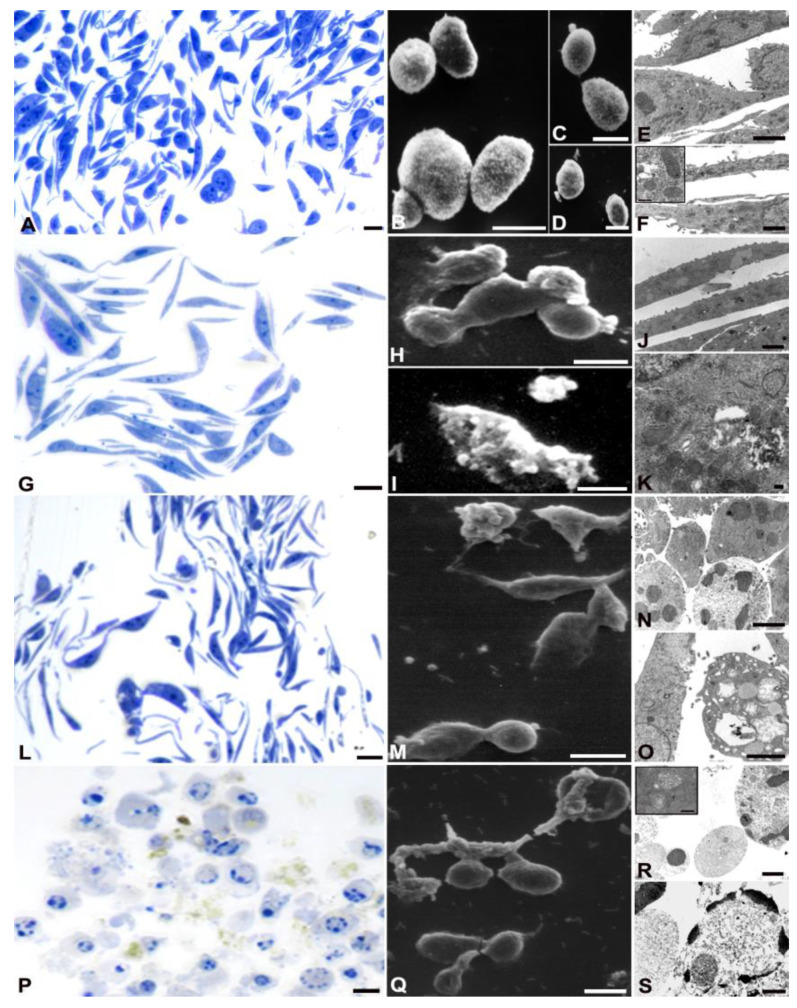
Optical images of cells stained with toluidine blue (**A**,**G**,**L**,**P**), SEM (**B**–**D**,**H**,**I**,**M**,**Q**) and TEM (**E**,**F**,**J**,**K**,**N**,**O**,**R**,**S**) analyses of AGS cells treated with different AOME doses. (**A**,**G**,**L**,**P**) Bar = 20 µm; (**B**–**D**,**H**,**I**,**M**,**Q**) Bar = 10 µm; (**E**,**F**,**J**,**K**,**N**,**O**,**R**,**S**) Bar = 2 µm.

**Figure 4 ijms-24-16003-f004:**
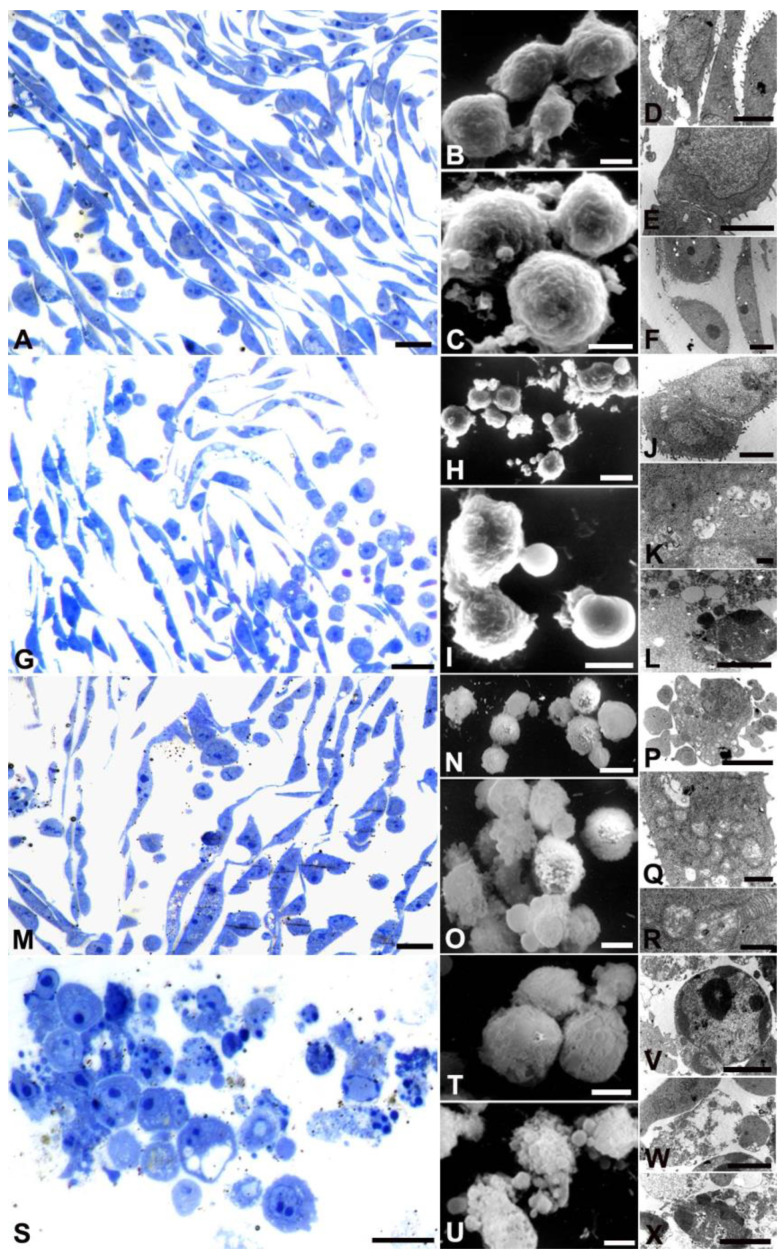
Images of cells stained with toluidine blue (**A**,**G**,**M**,**S**), SEM (**B**,**C**,**H**,**I**,**N**,**O**,**T**,**U**) and TEM (**D**–**F**,**J**–**L**,**P**–**R**,**V**–**X**) analyses of KATO III cells treated with different AOME doses. (**A**,**G**,**M**,**S**) Bar = 20 µm; (**B**,**C**,**H**,**I**,**N**,**O**,**T**,**U**) Bar = 10 µm; (**D**,**F**,**J**,**L**,**P**,**V**–**X**) Bar = 5 µm; (**E**) Bar = 2 µm; (**K**,**Q**,**R**) Bar = 1 µm.

**Figure 5 ijms-24-16003-f005:**
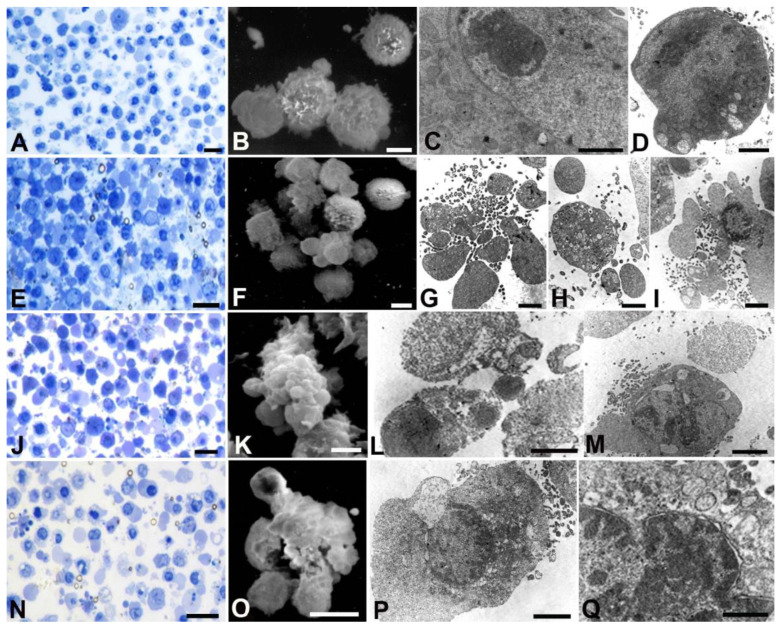
Optical (**A**,**E**,**J**,**N**), SEM (**B**,**F**,**K**,**O**) and TEM (**C**,**D**,**G**–**I**,**L**,**M**,**P**,**Q**) analyses of SNU 1 cells treated with different AOME doses. (**A**,**E**,**J**,**N**) Bar = 20 µm; (**B**,**F**,**K**,**O**) Bar = 5 µm; (**D**,**G**,**H**,**I**,**M**) Bar = 1 µm; (**C**,**L**,**P**,**Q**) Bar = 2 µm.

**Table 1 ijms-24-16003-t001:** Concentration (mg kg^−1^ of dried extract) of bioactive compounds in AOME.

No.	Compounds	Concentration mg kg^−1^
1	Gallic acid	17.07
2	Neochlorogenic acid	14.49
3	Delphindin-3-galactoside	19.85
4	(+)-Catechin	n.d.
5	Procyanidin B2	n.d.
6	Chlorogenic acid	44.52
7	*p*-Hydroxybenzoic acid	725.76
8	(−)-Epicatechin	n.d.
9	Cyanidin-3-glucoside	n.d.
10	Petunidin-3-glucoside	n.d.
11	3-Hydroxybenzoic acid	1160.66
12	Caffeic acid	9867.16
13	Vanillic acid	n.d.
14	Resveratrol	n.d.
15	Pelargonidin-3-glucoside	n.d.
16	Pelagonidin-3-rutinoside	n.d.
17	Malvidin-3-galactoside	n.d.
18	Syringic acid	n.d.
19	Procyanidin A2	n.d.
20	*p*-Coumaric acid	9976.83
21	Ferulic acid	2696.19
22	3,5-Dicaffeoylquinic acid	n.d.
23	Rutin	483.97
24	Hyperoside	2540.33
25	Isoquercitrin	986.27
26	Delphindin-3,5-diglucoside	867.66
27	Phloridzin	2.64
28	Quercitrin	n.d.
29	Myricetin	n.d.
30	Naringin	n.d.
31	Kaempferol-3-glucoside	342.52
32	Hesperidin	n.d.
33	Ellagic acid	428.46
34	*trans*-cinnamic acid	603.99
35	Quercetin	90.59
36	Phloretin	n.d.
37	Kaempferol	81.17
38	Isorhamnetin	5.73
Total content		30,956.87

n.d., not detectable. Relative standard deviation (RSD) for all compounds ranged from 3.51 to 9.14%.

## Data Availability

All data were generated in-house, and no paper mill was used. All authors agree to be accountable for all aspects of work ensuring integrity and accuracy.

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
