# Peer review of "Paradigm Shift in Gastric Cancer Prevention: Harnessing the Potential of Aristolochia olivieri Extract"

_ijms, 2023, doi:10.3390/ijms242116003_

Round 1
Reviewer 1 Report
Comments and Suggestions for Authors
Please provide full name of AOME at the first mention
Line 156: Notably, other flavonoids like (+)-catechin, procyanidin B2, (-)-epicatechin, cyanidin-3-gluco
side, petunidin-3-glucoside, quercitrin, myricetin, naringin, hesperidin, phloretin,
kaempferol, and isorhamnetin were not detected.
This sentence should be changed.First of all, procyanidin B2 is not a flavonoid, it is a condensed tannin. Furthermore, it better to give (+)-catechin and (-)-epicatechin as tannin precursors. Therefore, I suggest to change the sentence. At least ‘other flavonoids’ can be replaced by ‘other polyphenols’.
Line 79: It is better to write only northern Irak. I recommend to remove ‘Kurdistan region’. This
Figure 1: please use mg/mL or mg/ml. They should be in the same style
Author Response
Referee 1
(x) I would not like to sign my review report
( ) I would like to sign my review report
Quality of English Language
( ) I am not qualified to assess the quality of English in this paper
( ) English very difficult to understand/incomprehensible
( ) Extensive editing of English language required
( ) Moderate editing of English language required
( ) Minor editing of English language required
(x) English language fine. No issues detected
Yes Can be improved Must be improved Not applicable
Does the introduction provide sufficient background and include all relevant references?
(x) ( ) ( ) ( )
Are all the cited references relevant to the research?
(x) ( ) ( ) ( )
Is the research design appropriate?
( ) ( ) ( ) ( )
Are the methods adequately described?
(x) ( ) ( ) ( )
Are the results clearly presented?
(x) ( ) ( ) ( )
Are the conclusions supported by the results?
(x) ( ) ( ) ( )
Comments and Suggestions for Authors
Please provide full name of AOME at the first mention
- We thank the referee for this suggestion. In the subsection “purpose” of the abstract, it has been added the following expression “an Aristolochia olivieri Colleg. ex Boiss. leaves methanolic extract” before AOME
Line 156 became line 98: Notably, other flavonoids like (+)-catechin, procyanidin B2, (-)-epicatechin, cyanidin-3-glucoside, petunidin-3-glucoside, quercitrin, myricetin, naringin, hesperidin, phloretin, kaempferol, and isorhamnetin were not detected.
This sentence should be changed. First of all, procyanidin B2 is not a flavonoid, it is a condensed tannin. Furthermore, it better to give (+)-catechin and (-)-epicatechin as tannin precursors. Therefore, I suggest to change the sentence. At least ‘other flavonoids’ can be replaced by ‘other polyphenols’.
- This was the line 148 that became line 98. The word “flavonoids “was replaced with the word “polyphenols”
Line 79 became 231-232: It is better to write only northern Irak. I recommend to remove ‘Kurdistan region’. This
- It was line 231-232. As suggested, we removed “Kurdistan region”
Figure 1: please use mg/mL or mg/ml. They should be in the same style
- We use mg/ml in Figure 1, as suggested by reviewer.

Reviewer 2 Report
Comments and Suggestions for Authors
Author Response
Referee 2
1 - If abbreviations are used in the abstract, they must be explained, otherwise the reader may
be confused. Please explain what is the meaning of AOME in the abstract.
- We thank the referee for this suggestion. In the "purpose" subsection of the abstract, we have incorporated the following phrase: "an Aristolochia olivieri Colleg. ex Boiss. leaves methanolic extract" before 'AOME.'"
2 – The difference between the three cell lines used in the study should be mentioned in the text. I suggest to include this information in the section Material and methods – subsection Cell culture and treatments
- We thank the referee for this suggestion that can improve the manuscript clarity. We added “(ATCC-CRL1739)” after “AGS”, “(ATCC-CRL5971)” after “KATO III”, “(ATCC-HTB103)” after SNU-1. Furthermore we added the following text in order to explain the differences among the three cell lines used in the study “These cell lines represent distinct histological subtypes of gastric carcinoma. AGS models adenocarcinoma, KATO-III represents signet ring cell carcinoma, and SNU-1 corresponds to undifferentiated adenocarcinoma. They are widely employed in scientific research for their ability to cover the spectrum of gastric carcinoma histologies, making them valuable tools for validation studies in the field.”
3 – A figure legend should not present the results themselves, but rather mention the analysis
employed to obtain the results (see Figure 1 and rephrase the legend)
- “AOME treatment leads to decreased cell proliferation/viability in gastric cell lines. “was replaced with “AOME's impact on the viability of KATO III and AGS cell lines at 24 and 48 hours of incubation” as suggested by the referee.
4- The graphs in figure 2 must be made with another software (for example GraphPad, or other).
- We thank the referee for this suggestion. In order to better explain our findings, we added the following sentence in order to explain the statistical analyses performed.: “Viability graphs have been realized with Graph Pad Software and, for statistical analysis, t-test has been performed, evaluating control (ctrl) sample vs treated sample both at 24h and 48h of treatment. Data have been considered statistically significant with pvalue <0.05 (asterisk) and highly significant with pvalue < 0.005 (two asterisks).”
In addition, the graphs in figure 2 were realized with GraphPad as suggested by the reviewer
5 – I believe that at least the main results should be supported by a RT-PCR analysis evaluating
the BAX/Bcl-2 ratio.
- We are grateful for your thoughtful suggestion regarding the inclusion of RT-PCR analysis to evaluate the BAX/Bcl-2 ratio in our manuscript. Your feedback underscores the importance of a comprehensive assessment of our findings. We acknowledge and totally agree with the relevance of BAX/Bcl-2 ratio in understanding apoptosis and its implications in gastric cancer. While this study primarily focused on chemical characterization and morphological analyses, we are actively pursuing an extended research effort to investigate the underlying molecular mechanisms involved. In addition to RT-PCR analysis, we are also exploring the role of microRNAs (miRNAs) in the regulatory pathways associated with apoptosis and their potential implications in gastric cancer. Given the complexity of these molecular investigations, we have made a strategic decision to present the results of the molecular and miRNA analyses in a subsequent publication. This approach allows us to provide a comprehensive and detailed account of each aspect of our study, emphasizing both the chemical and morphological analyses. We sincerely appreciate your understanding of our approach and your interest in our work. We are committed to presenting a robust and complete body of research that encompasses all relevant aspects of our study. Thank you for your invaluable feedback and guidance.

Reviewer 3 Report
Comments and Suggestions for Authors
In this study, Micucci et al, investigated the potential of AOME in context of gastric cancer by characterizing the chemical composition of AOME and evaluating its in vitro antibacterial and antitumor activities. While the study is well-demonstrated, I do have some comments for the author:
- Author should provide a brief introduction about AOME and should mention why AOME was chosen for investigation, particularly in the context of its relevance to gastric cancer. Also write the full form of AOME in abstract and introduction at least one time.
- Authors should include statistical analysis and p-values to quantify the significance of the findings.
- Author should consider elaborating on the specific areas where further research is needed to explore the potential clinical applications of AOME.
- There are some minor grammatical and typographical errors. Author should carefully proofread the manuscript.
Minor editing required
Author Response
Open Review
( ) I would not like to sign my review report
(x) I would like to sign my review report
Quality of English Language
( ) I am not qualified to assess the quality of English in this paper
( ) English very difficult to understand/incomprehensible
( ) Extensive editing of English language required
( ) Moderate editing of English language required
(x) Minor editing of English language required
( ) English language fine. No issues detected
Yes Can be improved Must be improved Not applicable
Does the introduction provide sufficient background and include all relevant references?
( ) ( ) (x) ( )
Are all the cited references relevant to the research?
(x) ( ) ( ) ( )
Is the research design appropriate?
( ) (x) ( ) ( )
Are the methods adequately described?
(x) ( ) ( ) ( )
Are the results clearly presented?
( ) (x) ( ) ( )
Are the conclusions supported by the results?
( ) (x) ( ) ( )
Comments and Suggestions for Authors
In this study, Micucci et al, investigated the potential of AOME in context of gastric cancer by characterizing the chemical composition of AOME and evaluating its in vitro antibacterial and antitumor activities. While the study is well-demonstrated, I do have some comments for the author:
Author should provide a brief introduction about AOME and should mention why AOME was chosen for investigation, particularly in the context of its relevance to gastric cancer. Also write the full form of AOME in abstract and introduction at least one time.
- We thank the referee for this important suggestion. In the introduction part, in line 76, we added the following sentence “In our study, we chose Aristolochia olivieri ex Boiss. due to its historical use in Kurdish folk medicine for gastrointestinal ailments. We opted for a methanolic extract because methanol is an effective solvent for extracting phenolic acids and flavonoids, compounds with known potential against gastric cancer. This choice allowed us to comprehensively examine the plant's chemical composition and evaluate its anti-cancer properties through apoptosis, making it a well-suited candidate for addressing gastric cancer.”
- In addition, in the subsection “purpose” of the abstract, it has been added the following expression “an Aristolochia olivieri Colleg. ex Boiss. leaves methanolic extract “ before AOME
Authors should include statistical analysis and p-values to quantify the significance of the findings.
- At the end of the section Material and methods, we added the following explanation Viability graphs have been realized with Graph Pad Software and, for statistical analysis, t-test has been performed, evaluating control (ctrl) sample vs treated sample both at 24h and 48h of treatment. Data have been considered statistically significant with pvalue <0.05 (asterisk) and highly significant with pvalue < 0.005 (two asterisks)”
as suggested by the editor
- At the end of the antibacterial activity section, the following sentence was added: “The MIC (Minimum Inhibitory Concentration) and MBC (Minimum Bactericidal Concentration) values were determined based on three replicates of each test. These values were assessed descriptively, as the primary analysis focused on identifying the lowest concentration that inhibited bacterial growth (MIC) and the concentration that resulted in bacterial cell death (MBC) on established microbiological guidelines and the consistency of results across the repeated experiments.”
Author should consider elaborating on the specific areas where further research is needed to explore the potential clinical applications of AOME.
- We appreciate your feedback. Regarding the potential clinical applications of AOME, we have elaborated on its likely clinical space. We propose that AOME may be considered for administration to individuals with H. pylori infection and an elevated risk of gastric carcinoma, aligning with the concept of 'nutraceuticals before drugs beyond the foods, adding the following sentence in line 219 “Our preliminary findings suggest that AOME may hold promise in a distinct clinical space, which may not be primarily associated with its antibiotic effects, but rather with the administration of the phyto-complex to individuals who have contracted H. pylori infection and, as a result, are at a higher risk of developing gastric cancer. This perspective aligns seamlessly with the concept of "nutraceuticals before drugs beyond the foods.”
There are some minor grammatical and typographical errors. Author should carefully proofread the manuscript.
- The authors as suggest by reviewer carefully proofread the manuscript.
Thank you for your review and feedback. We appreciate your attention to detail. We want to inform you that we have already proofread the manuscript and made the necessary corrections to address the minor grammatical and typographical errors. Your input has been invaluable in enhancing the quality of our work.

Round 2
Reviewer 2 Report
Comments and Suggestions for Authors
The manuscript was sufficiently improved to warrant publication in IJMS
Comments on the Quality of English LanguageMinor english editing